# ToMA: Token Merge with Attention for Image Generation with Diffusion Models

## Abstract

Diffusion is one of the leading approaches for image generation. Plug-and-play token merge techniques have recently been introduced to mitigate the high computation cost of transformer blocks in diffusion models. However, existing methods overlook two key factors: (1) the token selection process fails to account for relationships among tokens, potentially discarding important information and limiting image quality; 2) they do not take advantage of the modern, efficient implementation of attention, so that, the overhead backfires the achieved algorithmic efficiency. In this paper, we propose Token Merge with Attention (ToMA) with three major improvements. Firstly, we utilize a submodular-based token selection method to identify diverse tokens as merge destinations, representative of the entire token set. Secondly, we use efficient attention implementation for the merge operation with negligible overhead. Also, we formalize the (un-)merge as (inverse-)linear transformations, allowing shareable computation across layers/iterations. Finally, we utilize the image locality to further accelerate the computation by performing all the operations on tokens in local tiles. ToMA achieves the best trade-offs between speed-ups and generation quality compared to the baselines.

## 1 Introduction

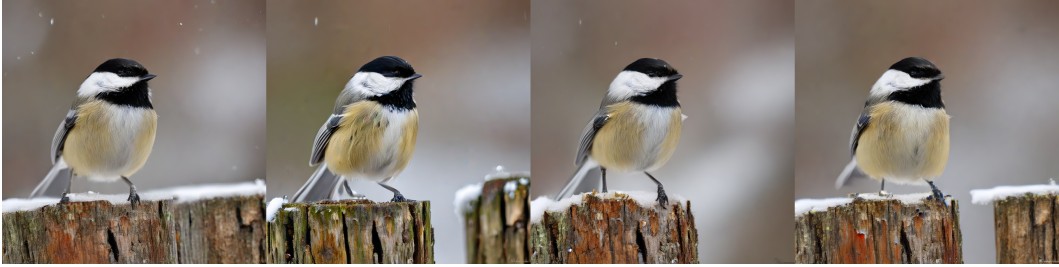

Figure 1: Variants of ToMA generated images: ToMA_stripe, ToMA, ToMA_tile, ToMA

Diffusion Ho et al. (2020); Song et al. (2021); Dhariwal & Nichol (2021) emerges as one of the leading approaches for high-quality image generation. However, the increasing complexity of diffusion models, driven by their core transformer-based architecture, presents significant computational challenges. The design of the transformer leads to quadratic complexity with respect to the number of tokens, making them inefficient and resource-intensive as token counts increase.

Methods with different approaches have been developed to mitigate this issue. Flash Attention Dao et al. (2022); Dao (2023) introduces a more efficient attention mechanism that reduces memory overhead, while xformers Lefaudeux et al. (2022) utilize sparse attention to lower memory usage and improve scalability. Methods like Token Pruning Kim et al. (2022) reduce computation by eliminating less relevant tokens during inference, albeit at the cost of potential quality degradation.

ToMeSD Bolya & Hoffman (2023) leverages token merging Bolya & Hoffman (2023); Kim et al. (2023), consolidating similar tokens during the forward pass to reduce the token count in computational layers, thereby lowering complexity without requiring network retraining. Essentially, the

original sequence of tokens gets merged before each layer in the transformer block, including attentions and MLPs, and after the computation finishes, an unmerge is applied to transform the merged tokens back to the original sequence length. Token merge shares the spirit with token pruning by reducing the input size to transformers, and is orthogonal to other acceleration methods such as Flash Attention and xformers.

Though ToMeSD has shown considerable theoretical speedups by significantly reducing the number of tokens, it struggles to accelerate diffusion models in practice with modern attention implementation and advanced GPU architectures. This is because the merge algorithm of ToMeSD Bolya & Hoffman (2023) requires relatively costly operations on GPU (e.g., sorting). This creates significant overhead that overshadows the speedups gained from token reduction, especially with more efficient implementations of attention (e.g., Flash Attention Dao (2023)) and GPU architectures better optimized for attention-like operations.

In this paper, we propose Token Merge with Attention (ToMA) to get practical speedups for diffusion models in a plug-and-play manner. Our method first utilizes a submodular function to identify a representative subset of tokens as merge destinations with a vectorized optimization algorithm that runs efficiently on GPUs. We then perform token merge by using an attention-like operation between the destination tokens and all tokens in the sequence, resulting in a linear transformation. For unmerge, we utilize the inverse or the transpose of such a linear transformation. The design of ToMA carefully considers the advantages and limitations of GPU computations.

To further reduce the overhead of ToMA, we leverage the locality characteristics of the hidden states within the latent space, which preserves image locality, so the tokens are more likely to be similar within a local region. By partitioning the hidden states into local regions, we can run ToMA in each region independently and mitigate the overhead costs by reducing the input size to ToMA while enjoying the parallelism of computation. Moreover, we also find that the destination selection and linear transformation of merge and unmerge can be shared across network layers and diffusion steps, which further decreases the ToMA overhead costs. As a result, ToMA achieves 30%-50% speedups without noticeable sacrifice in image quality.

## 2 RELATED WORK

**Efficient Transformer:** The core of Transformers' Vaswani et al. (2017) quadratic time complexity poses a bottleneck for both inference and training. Various methods have attempted to address this problem. To reduce computation complexity, ReformerKitaev et al. (2020) uses locality-sensitive hashing where LinformerWang et al. (2020), PerformerChoromanski et al. (2020), Low-Rank TransformerWinata et al. (2020) leverage low-rank approximations of the self-attention matrix to speed up computation. PerceiverJaegle et al. (2021), CharformerTay et al. (2021), and Funnel TransformersDai et al. (2020) use different ways to downsample the input to reduce the computation cost. Moreover, Sparse TransformerChild et al. (2019) and Big BirdZaheer et al. (2020) design different sparse attention patterns to let each token attend to a subset of all tokens. FRDiff So et al. (2024) accelerates diffusion inference by reusing feature maps across time steps but not merging the tokens.

**Learned Token Reduction** The majority of learned token reduction involve training auxiliary models to assess the importance of tokens in the input data. For example, DynamicViT Rao et al. (2021) employs a lightweight MLP module to generate pruning masks based on input token features. These masks are learned through a distillation process. GQA Ainslie et al. (2023) introduces an innovative mechanism that shares key and value heads across multiple query heads, balancing between the flexibility of multi-head attention and the efficiency of multi-query attention. A-ViT Yin et al. (2022) efficiently computes halting probabilities using the first channel of features, guided by auxiliary losses. Despite their effectiveness, these methods often require additional fine-tuning of auxiliary modules, which can be seen as a limitation. **Language-Vision Acceleration**. CrossGET Shi et al. (2024) combines token but on vision-language models with tasks like image captioning and image-text retrieval. TRIPS Ye et al. (2024) proposes text-relevant image patch selection but it accelerates the image-language model pertaining. DiffRate Chen et al. (2023) incorporates the compression rate and merges tokens in the vision transformers but in the training stage.

**Heuristic Token Reduction** Unlike learned token reduction techniques, some works have introduced heuristic token reduction strategies that can be directly applied to pre-trained ViTs without

requiring additional fine-tuning. For instance, Adaptive-Token Sampling Fayyaz et al. (2022) selects tokens based on their similarity to the class token in the attention map, which outperforms the top-k sampling. However, the requirement of the class token poses a limitation in dense prediction tasks such as image generation. Token Pooling Marin et al. (2021) merges spatially adjacent tokens within a local window to reduce the token count at various stages of the ViT. Token merge Bolya et al. (2023) introduced a different pooling method that merges similar tokens based on an efficient bipartite matching algorithm. ToMeSD Bolya & Hoffman (2023) randomly groups tokens into source and destination groups and merges the source tokens with the destination tokens based on the pair similarity score. This is one of our baselines.

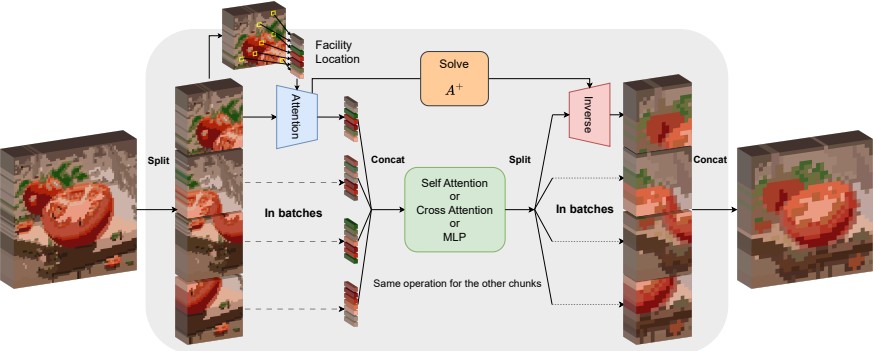

Figure 2: Overview of ToMA. **Facility Location** selects representative destination tokens $D$ from the token set $N$ using submodular optimization. **Attention** computes a low-rank projection matrix mapping $N$ to $D$, followed by standard transformer operations (e.g., self-attention, cross-attention, or MLP) on $D$. **Inverse** reconstructs $N$ from $D$ via a pseudo-inverse or transpose. These steps can be applied locally to latent space regions as batch operations for efficiency.

## 3 PRELIMINARIES

**Attention Computation and Notation.** We denote the attention computation as SDPA(scaled dot product attention). The input query is $\boldsymbol{Q} \in \mathbb{R}^{N \times d}$, key is $\boldsymbol{K} \in \mathbb{R}^{N \times d}$, and value is $\boldsymbol{V} \in \mathbb{R}^{N \times d}$. $N$ is sequence length and $d$ is the hidden state dimension.

$$\text{SDPA}(\boldsymbol{Q}, \boldsymbol{K}, \boldsymbol{V}) = \text{softmax}\left(\frac{\boldsymbol{Q}\boldsymbol{K}^T}{\sqrt{d}}\right)\boldsymbol{V} \tag{1}$$

Additionally, we denote $D$ as the size of all destination tokens and $B$ as the batch size, $\boldsymbol{X}$ as the latent matrix of shape $N \times d$ that gets projected into $\boldsymbol{Q}$, $\boldsymbol{K}$ and $\boldsymbol{V}$ (for simplicity, we assume the same feature dimension in latents and attention). Matrices and vectors are in **bold**, while others are not.

**Submodularity.** A submodular function (Fujishige, 2005) is a set function $f : 2^V \rightarrow \mathbb{R}$ with the diminishing return property: $f(v|A) \geq f(v|B)$ if $v \notin B$, $A \subseteq B$, where $f(v|A) := f(v \cup A) - f(A)$. Intuitively, the property states that the gain of a smaller subset is always greater or equal to that of a larger subset. This makes submodular function $f(A)$ very useful in expressing the diversity of the input subset $A$ relative to the ground set $V$.

**for** $i = 1 \ldots k$ **do**
$\quad v^* \in \arg\max_{v' \in V \setminus A} f(v'|A)$;
$\quad A = A \cup \{v^*\}$;
**end**
**return** $A$

**Algorithm 1** Greedy

The submodular maximization problem with a cardinality constraint is shown below 2.

$$\max_{A \subseteq V} f(A) \quad \text{s.t. } |A| \leq k \tag{2}$$

The greedy algorithm (Alg. 1) guarantees a $(1 - 1/e)$-approximation of the optimal solution. It iteratively selects the element that maximizes the gain until the chosen set size reaches $k$.

**Conditional Diffusion Model.** The conditional diffusion model is a variation from the diffusion model. Different from the unconditional diffusion model, the conditional one estimates the data distribution with the additional information. The forward noise process is defined in 3

$$q(\boldsymbol{x}_t|\boldsymbol{x}_0) \coloneqq \mathcal{N}(\boldsymbol{x}_t|\sqrt{\bar{\alpha}_t}\boldsymbol{x}_0, (1-\bar{\alpha}_t)\boldsymbol{I}), \tag{3}$$

The model gradually adds noise to the input image over steps $t$ to transform input $\boldsymbol{x}_0$ to a latent noise representation. $\bar{\alpha}_t \coloneqq \prod_{s=0}^{t} \alpha_s = \prod_{s=0}^{t}(1-\beta_s)$ and $\beta_s$ represents the noise variance schedule Ho et al. (2020). The denoising process below

$$p_\theta(\mathbf{x}_{t-1}|\mathbf{x}_t) = \mathcal{N}(\mathbf{x}_{t-1}; \boldsymbol{\mu}_\theta(\mathbf{x}_t, t), \boldsymbol{\sigma}_t^2\boldsymbol{I}) \tag{4}$$

is parameterized by the neural network $\mu_\theta$, where $\boldsymbol{\sigma}_t^2$ denotes the transition variance. It aims to iteratively reconstruct $\boldsymbol{x}_0$ from the random noise.

# 4 ToMA

Token merge selects several destination tokens from the full set. It merges the tokens into destinations based on similarity scores, typically by assigning the merged destination as the average of the merge tokens. If the merged tokens are very similar, e.g., regions of pixels that represent the background with homogeneous colors, then the loss of information in the merge process can be minimized. Compared to token pruning, which directly throws away tokens, the token merge retains more information, thus achieving better image quality.

Token merge reduces the number of inputs processed in the transformer block, leading to significant computational savings. Thus, we can achieve a theoretical speedup based on the token merge ratio and the computational complexity of the transformer block (details in Appendix). In the unmerge process, the values of the merged tokens are redistributed back to their original tokens to reverse the merge process. This operation ensures that the information from the merged tokens is restored while maintaining the shape of the output without token merge so that later layers can process without any modifications.

ToMA consists of three key stages, where we achieve significant improvements: 1) Destination Token Selection: Efficiently selecting the most representative tokens as destinations. 2) merge Tokens: merge source tokens into their corresponding destination tokens based on similarity computed using attention. 3) Unmerge Tokens: Restoring the merged tokens to their original forms by reversing the merge process as a linear operation. We also get further speedups by a) utilizing the locality characteristics of the latent space and b) sharing destination/merge/unmerge computations across iterations and layers to reduce overhead.

## 4.1 Submodular-Based Destination Selection

Let $\boldsymbol{S}$ be the cosine similarity matrix between all token hidden representations. $\boldsymbol{S}_i$ represents the $i$-th row of the $\boldsymbol{S}$ matrix, and $\boldsymbol{S}_{i,j} \coloneqq \cos(\boldsymbol{X}_i, \boldsymbol{X}_j)$. We denote the chosen destination token set as $T$, and all the tokens (the ground set in submodular optimization) as $V$. $L \in \mathbb{R}^{|V|}$ is the max cache vector and $L_i$ is the max similarity score between the token $i$ in the ground set to our chosen set $T$.

The submodular function we use for destination selection is the facility location function (FL) $f_{\text{FL}}$ shown in Eq. 5. FL sums the similarity between every token $v_i \in V$ with its most similar neighbor in the selected destination set $v_j \in T$. Therefore, a high value for $f_{\text{FL}}(T)$ means every token $v_i$ has a similar neighbor in $T$, and $T$ is representative of $V$, which perfectly matches the goal of merge destination selection. We also note that our framework is general, and $f_{\text{FL}}$ could be potentially replaced with any other submodular function.

$$f_{\text{FL}}(T) = \sum_{v_i \in V} \max_{v_j \in T} \boldsymbol{S}_{i,j} \tag{5}$$

When optimizing $T$ using the greedy algorithm, we essentially need to identify the next best token with the largest gain $f_{\text{FL}}(v|T')$ relative to the so-far-selected set $T'$. The largest gain can be

decomposed (details in Appendix) as $\arg \max_{i \notin T'} \sum_{j=1}^{N} max(0, \boldsymbol{S}_{i,j} - \boldsymbol{c}_j(T'))$ with $\boldsymbol{c}$ as a vector containing the cached max values of $T'$ and updated incrementally as we select the next destination token: $\boldsymbol{c}_j(T') = \max_{v_l \in T'} \boldsymbol{S}_{j,l}$. We can perform all those operations in matrix forms, which makes them a perfect fit for GPU computation. There are more efficient submodular optimization algorithms compared to greedy, such as lazier-than-lazy greedy (Mirzasoleiman et al., 2015). However, the more complicated algorithms introduce operations (e.g., random subset selection) that are not ideal for GPU implementations.

### 4.1.1 WHY SUBMODULAR

The submodular function, particularly the facility location function, offers a theoretical guarantee for optimizing the selection of elements with the highest information gain from a set. This characteristic aligns well with our requirements for token selection, ensuring that we choose the most similar tokens for merge with minimal information loss. Furthermore, facility location is highly compatible with GPU implementations, as it takes advantage of matrix operations, significantly boosting computational efficiency. Finally, facility location is compatible with an arbitrary similarity function, where we use cosine similarity that more closely aligns with the attention computation (supposing the input tokens are properly normalized using, e.g., Layernorm (Ba, 2016)).

We have also considered clustering-based methods such as k-means for token selection. However, we opted against them for several reasons. First, k-means provides a soft target, which might introduce artifacts and achieve suboptimal performance. Second, k-means assumes that clusters have ball-like boundaries, which is not flexible and imposes extra assumptions on the latent space. Third, the k-means method requires variable iterations to converge, and it is hard to control the computational costs and trade-off with the clustering quality.

## 4.2 MERGE AND UNMERGE WITH ATTENTION

In ToMA, we achieve token merge through a linear transformation approach that uses attention weights to assign tokens. This allows for a more generalized merge process. Accordingly, the unmerge operation can be the inverse of the linear transformation.

### 4.2.1 MERGE

We first compute softmax similarity weights between all destination tokens and all source tokens. This weight can be optionally sharpened or softened using the temperature parameter. Next, we normalize the matrix by counting the sum of each row and dividing the corresponding row by that value. Finally, we merge tokens together as a weighted average.

$$\boldsymbol{A} = \text{SDPA}(\boldsymbol{X}_T, \boldsymbol{X}, \boldsymbol{I}, \tau) = \text{softmax}\left(\frac{\boldsymbol{X}_T \boldsymbol{X}^\top}{\tau}\right) \boldsymbol{I}$$

$$\tilde{\boldsymbol{A}}_{ij} = \frac{\boldsymbol{A}_{ij}}{\sum_j \boldsymbol{A}_{ij}} \quad \text{(Normalize each row of } \boldsymbol{A}\text{)}, \ \boldsymbol{X}_{\text{merged}} = \tilde{\boldsymbol{A}} \boldsymbol{X} \tag{6}$$

We describe the merge operation in Eq. 6. Here, $\boldsymbol{X}_T$ is the hidden representation matrix for the set of destinations (shape $D \times d$), and $\tau$ is the temperature. $\boldsymbol{X}_T$ are essentially sub-rows of $\boldsymbol{X}$ so that the attention is between the destinations and all the tokens. The softmax is computed over all the destinations for every source token, where intuitively, we can think every source token gets distributed to some destinations, and the sum of the weights is $1$. Because the source tokens include the destinations, in the worst case, every destination gets assigned by itself (e.g., if the destination is dissimilar to all other tokens). Note that we include the identity matrix to match the attention notation, which can be ignored in implementation.

For extremely small temperature values, the attention linear projection $\boldsymbol{A}$ contains 1's and 0's, so our attention-based merge recovers the hard discrete merge by approximating the average of the merged tokens. Moreover, as we essentially compute an attention matrix and use it as a linear projection on the source tokens, our merge can be highly efficient on modern GPU architecture. Also, the linear transformation can be stored and reused later.

### 4.2.2 UNMERGE

To unmerge tokens, we inverse the projection matrix of merge with the following two options (Eq. 7):

**Transpose** of the merge matrix $\boldsymbol{A}^{\top}$: By multiplying the transpose of the merge matrix with the output of the transformer block, we distribute the merged token values back to their original tokens. When the temperature is extremely low, the transpose unmerge copies the computation result from the destination token to the corresponding merged tokens. This incurs very little overhead.

**Pseudo-inverse** of the merge matrix $\boldsymbol{A}^{\dagger}$: Viewing the merge as a linear transformation, the pseudo-inverse minimizes the reconstruction error if the computation between merge and unmerge is close to linear. It is much more computationally expensive than $\boldsymbol{A}^{\top}$ and requires SVD or QR decomposition.

$$\boldsymbol{X}'_{\text{unmerged}} = \boldsymbol{A}^{\top}\boldsymbol{X}' \text{ or } \quad \boldsymbol{X}'_{\text{unmerged}} = \boldsymbol{A}^{\dagger}\boldsymbol{X}' = \boldsymbol{A}^{\top}(\boldsymbol{A}\boldsymbol{A}^{\top})^{-1}\boldsymbol{X}' \tag{7}$$

$\boldsymbol{A}^{\top}$ and $\boldsymbol{A}^{\dagger}$ are the same if the rows of the merge matrix $\boldsymbol{A}$ are independent, e.g., source tokens are not overlapping among different destinations. Intuitively, this means that the destinations should be as diverse as possible, which also matches the objective of the submodular optimization. Also, when the temperature is extremely low, every source token gets assigned to a single destination token, so the two options are identical. Concerning efficiency, we opt to use the transpose as the default unmerge method for ToMA.

### 4.3 FURTHER SPEEDUP

The overhead of ToMA consists of 1) the computation of destination tokens using submodular optimization, 2) the computation of the attention merge and unmerge matrix, and 3) applying the merge and unmerge matrices before and after layers in the transformer block. We further reduce all three overheads by considering the locality of the feature space so we can perform the computations locally in every region. We also decrease the frequency to compute 1) and 2) by sharing destinations and merge matrices across iterations and layers.

### 4.3.1 LOCAL REGION

A crucial aspect ignored in ToMe for SD is the locality of the latent space (Fig. 3). The spatial relationship between the generated image and the hidden states within the UNet model becomes evident when examining this figure. We apply K-means to the tokens and recolor them based on their class affiliation. Specifically, by projecting the color from the generated image onto the hidden state feature map, we observe clear spatial coherence. The tokens in the latent space consistently demonstrate the greatest similarity with their neighboring tokens, creating distinct localized regions.

This observation aligns with the intuition that images exhibit local consistency and smoothness. Therefore, we hypothesize that the most likely merge destination for any token resides within its local tile. This allows us to focus exclusively on the near tokens token while ignoring more distant ones.

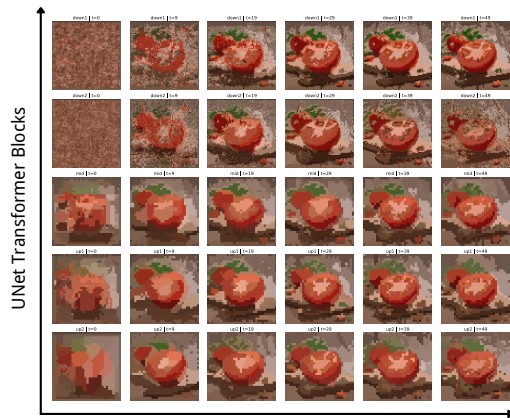

Figure 3: Recolored K-means results on UNet hidden states, across blocks/ denoising steps.

To exploit this property, we limit operations to local regions, performing token selection and (un)merge within each region. We propose two region selections:

**Tile-shape region**: The tile region approach is particularly effective because it comprehensively captures the local characteristics by considering the image's locality.

**Stripe-shape region**: The stripe region focuses on tokens on the same row, which misses the proximity in the vertical direction.

Both locality options can significantly reduce the computational overhead as we perform all operations in ToMA with a smaller number of input tokens in parallel. The tile-shape region is more coherent with the nature of the image and we find it performs better in experiments. However, turning a 2D matrix into tile-shape regions requires reshuffling, which brings additional overhead on GPUs. On the contrary, the stripe-shape option is faster as it only requires re-shaping of the 2D matrix while keeping its contiguous memory layout intact. We include both options in ToMA to provide trade-offs between speed and quality. Also, note that the tile-shape computation can be potentially accelerated as the low-level GPU operations are all in tiles, but it would require substantial re-implementation of the attention kernel. We defer this to future work.

We want to emphasize that the local region only affects components of ToMA, the computations in the transformer blocks always operate on the $D \times d$ matrix of the merged destinations. Please refer to Appendix Alg. 3 for the detailed algorithm of ToMA with local regions.

### 4.3.2 SHARING OVERHEAD COMPUTATION

Our observations of the diffusion denoising process reveal that hidden states show substantial similarity across steps, meaning the selected destinations are also quite alike. As shown in Fig. 4, destination tokens have a significant intersection with the ones chosen in previous steps. Therefore, we can share destination selections across steps. Additionally, the linear (un)merge operations enable us to reuse the matrices across layers with minimal quality loss. By sharing both destinations and attention weights across steps and layers, we significantly reduce the number of times for destination computation and attend merge matrix computation while maintaining high output quality.

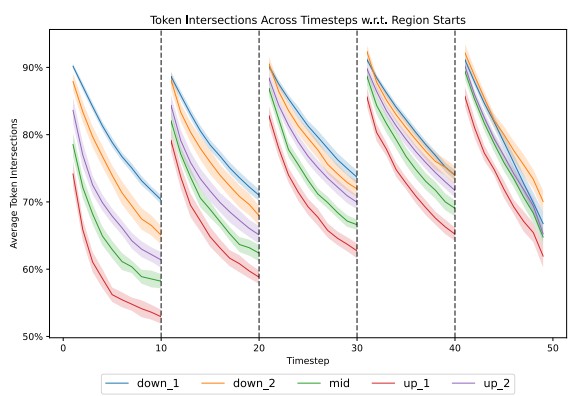

Figure 4: Intersection percentage of selected tokens between each step and the first step of its corresponding 10-step interval. Different curves refer to different layers in SDXL.

## 5 EXPERIMENTS

We evaluated ToMA on the SDXL stable-diffusion-xl-base-1.0 model using the Diffusers framework to generate $1024 \times 1024$ images. The prompts were sourced from the GEMRec dataset (Guo et al., 2024) and ImageNet_1K (Deng et al., 2009). To assess image quality, we used three primary metrics: CLIP, DINO, and FID (Radford et al., 2021; Caron et al., 2021; Heusel et al., 2017). For the CLIP and DINO evaluations, we generated images using 50 different prompts, each with 3 distinct seeds, and calculated the average score across all prompts and seeds. For measuring inference time, we reported the lowest wall-clock time over 100 runs.

**Diffusion Models**. We focus on the Stable Diffusion XL with the checkpoint of stable-diffusion-xl-base-1.0. SDXL is capable of generating very high-quality images and is popular in the community with abundant LoRAs available. Note that ToMA can generally apply to any transformer architecture. Thus, we can simply extend ToMA to other diffusion models like SD3 and SD2.

**Baseline**. ToMeSD (Bolya & Hoffman, 2023) selects tokens either in fixed or random small tiles, using a rigid approach. ToMeSD then discretely merges tokens by recording token pairs based on their computed similarity. In the unmerge phase, ToMeSD restores the original tokens by copying the values of the merged tokens back to their corresponding original tokens. We note that ToFu (Kim et al., 2023) is another relevant method that dynamically selects whether to prune or merge tokens based on the function's linearity to accelerate diffusion models. The work done by ToFu is orthogo-

nal to our research, meaning our methods can be seamlessly integrated with ToFu to further enhance its performance. Therefore, we don't include ToFu in our comparison.

## 5.1 RESULTS ON QUALITY AND EFFICIENCY

The primary objective of this comprehensive experiment is to evaluate the trade-off between the quality of generated images and the computational efficiency across different token merge methods. Specifically, we compare **ToMA** with **ToMeSD**, using three key metrics: **CLIP**, **DINO**, and **FID**. These metrics measure visual similarity, attention mechanisms, and image fidelity, respectively. Additionally, we assess the generation time and speed-up ratio for each method, offering insights into computational gains.

We introduce three versions of ToMA to explore the impact of different attention mechanisms and facility location strategies:

**ToMA(Stripe Facility Location + Global Attention)**: Combines stripe-based facility location with global attention for token merge.

**ToMA_stripe (Stripe Facility Location + Stripe Attention)**: Utilizes both stripe-based facility location and stripe attention for localized merge.

**ToMA_tile (Tile Facility Location + Tile Attention)**: Applies tile-based facility location and attention for tile-wise merge.

For the Stripe Attention and Tile Attention versions, a tile size of 16 is used, while the Global Attention version uses 256 tiles. We compute facility location every 10 steps and attention weights every 5 steps over a total of 50 steps. The ToMeSD method serves as a baseline for comparison across all versions, using the same CLIP, DINO, and FID metrics.

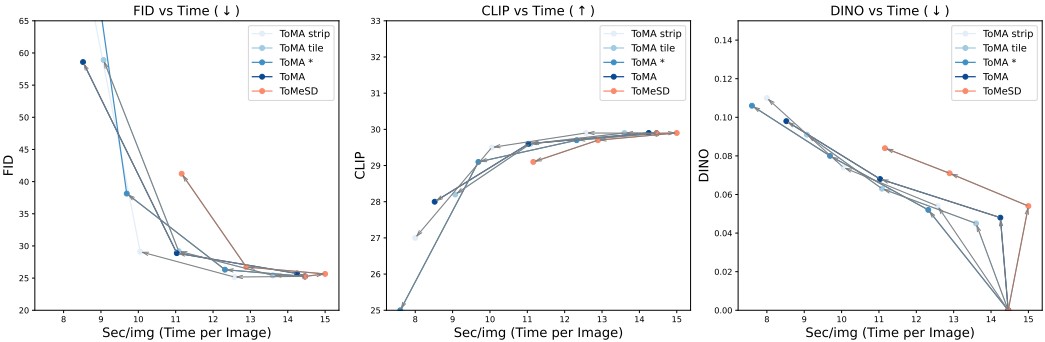

Figure 5: Quality metrics vs. generation time for `SDXL-base` on Nvidia V100. The merge ratio progresses from 0 to 0.75, moving from right to left following the directions of the arrows. Metrics are denoted as (↑: higher is better, ↓: lower is better).

In most cases, ToMeSD either increases computation costs or achieves minor speed-ups. Both versions of ToMA—stripe facility location with global attention and stripe facility location with stripe attention—maintain high image quality while delivering significant speed-ups. Tile facility location with tile attention achieves the best image quality, but the overhead is substantial, indicating great potential for further improvement through optimized low-level implementations.

In addition to evaluating image quality, we examine the generation time and speed-up ratio more variances. The token merge methods include ToMeSD, ToMA (with stripe and tile variations), and ToMA*, which employs a "merge-once" strategy (merge and unmerge once for the entire transformer block instead of applying ToMA on every component of the transformer individually). Furthermore, we compare these methods to the LB (lower bound for speed-up), which is the best speedup we can get with a linear project merge and unmerge approach (apply random merge and unmerge projections without other overhead computations).

This comprehensive experiment, conducted under the same experimental setup (including dataset, tile size, and step scheduler), allows us to test the trade-off between **time** and **quality** across different token merge methods.

| Token Merge Method | Generation Time / Speed Up Ratio | | |
|---|---|---|---|
| | **0.25** | **0.5** | **0.75** |
| **Baseline (ratio=0)** | 6.07s / 0.0% | 6.07s / 0.0% | 6.07s / 0.0% |
| **ToMeSD** | 8.66s / +42.7% | 8.73s / +43.8% | 8.16s / +34.4% |
| **ToMA** | 6.03s / -0.7% | 5.04s / -17.0% | **4.34s / -28.5%** |
| **ToMA_stripe** | 5.56s / -8.4% | **4.62s / -23.9%** | 4.48s / -26.2% |
| **ToMA_tile** | 6.20s / +2.1% | 6.27s / +3.3% | 6.23s / +2.6% |
| **ToMA*** | **5.45s / -10.2%** | 4.91s / -19.1% | 4.87s / -19.8% |
| **LB** | 5.16s / -15.0% | 4.01s / -33.9% | 3.13s / -48.4% |

Table 1: Comparison of Token Merge Methods and their Generation Time / Speed Up Ratios (RTX6000Ada). Negative percentages indicate faster times than the baseline, while positive percentages indicate slower times.

From Tab. 1, we find that ToMeSD exhibits increased generation times and negative speed-up ratios compared to the baseline across all token reduction ratios, ranging from +34.4% to +43.8%. This indicates that ToMeSD actually adds computational overhead rather than improving speed, making it less efficient than the baseline. We find that all ToMA variations, except ToMA_tile, deliver significant speed improvements over ToMeSD ranging from -28.5% to -0.7% . Moreover, methods like ToMA* and the ToMA stripe variants approach the theoretical speed limit, showcasing remarkable computational efficiency gains.

## 5.2 ABALATION TEST

In this section, we demonstrate on how we decide our default setting and other variances by comparing the image quality or speeds of combinations of different units in token merge.

### 5.2.1 FACILITY LOCATION & TILE FOR DESTINATION SELECTION

| Destination Selection | CLIP | DINO | Time (s) |
|---|---|---|---|
| Global Facility | 30.9486 | 0.0688 | 33.2 |
| Tile Facility | **31.0185** | **0.0550** | **5.1** |
| Stripe Facility | 30.9861 | 0.0740 | 5.16 |
| Random | 30.5527 | 0.0904 | 4.55 |

| num_tiles | CLIP | DINO | MSE | sec/img |
|---|---|---|---|---|
| **4** | 30.7747 | 0.0690 | 1564.0227 | 11.36 |
| **16** | 30.9914 | 0.0566 | 1345.3114 | 6.44 |
| **64** | 31.0185 | **0.0550** | **1274.1736** | 5.04 |
| **256** | **31.0273** | 0.0569 | 1296.3734 | **5.01** |

Table 2: Comparison of generated image metrics using different destination selection methods.

Table 3: Tile facility comparison with 50 recompute steps, ratio=0.5, global attention

From Tab. 2, we find that the facility location demonstrates great performance in the generated image metric, which proves our theory that we should find the most representative tokens during selection. Also, the tile facility achieves the best CLIP and DINO score which aligns with our observation of the hidden states locality. By restricting the token merge process in a local region, we get a better image quality. Thus, we utilize the tile facility as our default setting for ToMA.

From Tab. 3, we examine the influence of different tile sizes. We find that the tile sizes of 64 achieve the best score in DINO and MSE while 256 shows great performance in CLIP and time. Generally, the metric difference is not significant between these two tile sizes. Thus, we select 256 as our default setting due to its lead in speed. We report ablation results on comparison between transpose and pseudo-inverse as well as different sharing schedules in the Appendix.

## 5.3 MERGE AND UNMERGE SPEED

In this section, we compare the speed ToMA (un)merge which generalizes this process as linear transformation, and the discrete (un)merge of ToMeSD. In this experiment, we utilize the transpose strategy as the unmerge of the merge matrix.

| $N = 4096$ | 0.25 | 0.5 | 0.75 |
|---|---|---|---|
| **ToMeSD Merge** | 0.2107 | 0.2048 | 0.2028 |
| **ToMA Merge** | **0.0437** | **0.0403** | **0.0421** |
| **ToMeSD Unmerge** | 0.1607 | 0.1811 | 0.1579 |
| **ToMA Unmerge** | **0.0399** | **0.0483** | **0.0451** |

Table 4: Comparison of ToMeSD and ToMA speeds for size 4096.

| $N = 1024$ | 0.25 | 0.5 | 0.75 |
|---|---|---|---|
| **ToMeSD Merge** | 0.2022 | 0.2021 | 0.1932 |
| **ToMA Merge** | **0.0390** | **0.0388** | **0.0388** |
| **ToMeSD Unmerge** | 0.1605 | 0.1601 | 0.1440 |
| **ToMA Unmerge** | **0.0402** | **0.0405** | **0.0396** |

Table 5: Comparison of ToMeSD and ToMA speeds for size 1024.

Tab. 4 and Tab. 5 clearly demonstrate that the ToMA method significantly outperforms ToMeSD in both merge and unmerge speeds across all token reduction ratios (0.25, 0.5, and 0.75). For size 4096, ToMA achieves merge speeds approximately 80% faster than ToMeSD, with the lowest recorded merge time being 0.0403s compared to 0.2048s for ToMeSD. Similarly, the unmerge times for ToMA are consistently lower, with improvements ranging from 72% to 75% across the different token reduction ratios. This trend is mirrored in the 1024 size table, where ToMA again demonstrates its advantage, with merge and unmerge times consistently around 80% faster than ToMeSD. These results highlight the clear efficiency gains of the ToMA method in terms of both merge and unmerge processes, making it a more computationally efficient solution.

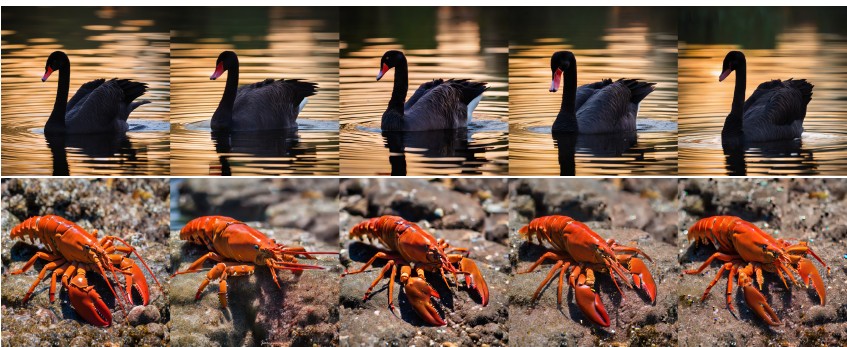

Figure 6: Image generated from left to right: original, ToMA_stripe, ToMA*, ToMA_tile, ToMA

### 5.4 DISCUSSION

Although the combination of tile facility location and tile merge produces high-quality images, it still falls short in speed. Optimizing this operation at a lower level could significantly reduce computational costs. Additionally, improving the implementation of the pseudo-inverse API would allow us to apply it to larger matrices, potentially enhancing image quality. Moreover, we utilize linear transformation, specifically SDPA, for token merge, where the parameter $V$ is currently set as an identity matrix and ignored during inference. This $V$ matrix holds potential for future training, which could further boost image quality.

**Broader impact**. ToMA enables speed improvements across a wide range of GPU architectures. On one hand, it accelerates image generation without compromising quality, making the process more efficient. On the other hand, it broadens the accessibility of diffusion models, allowing even those with less powerful or outdated GPUs to benefit from advanced techniques. ToMA reduces computational demands, making high-quality image generation feasible on a wider variety of hardware, thus making diffusion models more accessible to a larger audience.

## 6 CONCLUSION

In this work, we propose ToMA to enhance the existing token merge method in three key areas: 1) more representative token selection, 2) a more flexible and efficient merge and unmerge operation, and 3) the introduction of locality and sharing strategies. As a result, we achieve significant speedup while maintaining high image quality. For future work, we aim to further speed up the low-level tile region computation as well as fine-tune the merge attention for better generation quality.

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

# A   DETAILS ABOUT FACILITY LOCATION OPTIMIZATION

## A.1   GAIN FUNCTION COMPUTATION IN FL

In the greedy algorithm we select the token $v^*$ from $V \setminus A$ that maximizes the following gain function:

$$v^* = \arg\max_{v \in (V-A)} f(v|A)$$

where $f(v|A)$ is defined as:

$$f(v^*|A) = f(\{v^*\} \cup A) - f(A)$$
$$= \sum_{v \in V} \max_{v' \in (\{v^*\} \cup A)} \text{sim}(v, v') - \sum_{v \in V} \max_{v'' \in A} \text{sim}(v, v'')$$

Since for each $v \in V$, find the maximum corresponding $v'$ in the updated representatative set $\{v^*\} \cup A$ is equivalent to compare $v'$ in $A$ and $v^*$, namely:

$$\sum_{v \in V} \max_{v' \in (\{v^*\} \cup A)} \text{sim}(v, v') = \sum_{v \in V} max \left[ \left( \max_{v' \in A} \text{sim}(v, v'), \text{sim}(v, v^*) \right) \right]$$

Therefore

$$f(v^*|A) = \sum_{v \in V} \max \left[ \left( \max_{v' \in A} \text{sim}(v, v'), \text{sim}(v, v^*) \right) \right] - \sum_{v \in V} \max_{v'' \in A} \text{sim}(v, v'')$$
$$= \sum_{v \in V} \max \left[ \left( \max_{v' \in A} \text{sim}(v, v'), \text{sim}(v, v^*) \right) - \max_{v'' \in A} \text{sim}(v, v'') \right]$$
$$= \sum_{v \in V} \max \left( 0, \text{sim}(v, v^*) - \max_{v'' \in A} \text{sim}(v, v'') \right)$$

Eventually,

$$v^* = \arg\max_{v' \in (V-A)} \sum_{v \in V} \max \left( 0, \text{sim}(v', v^*) - \max_{v'' \in A} \text{sim}(v, v'') \right)$$

## A.2 FACILITY LOCATION OPTIMIZATION ALGORITHM

---

**Algorithm 2:** Facility Location Token Selection Algorithm

---

**Input:** Similarity matrix $S \in \mathbb{R}^{N \times N}$, number of tokens to select $D$
**Output:** Selected token indices $T$
**Initialize**: $T \leftarrow \{\}$;
**for** $i = 1$ **to** $N$ **do**
 $\quad$ Compute row sums: $s_i = \sum_{j=1}^{N} S_{i,j}$;
**end**
Select the first token: $t_1 \leftarrow \arg\max_i s_i$;
Add $t_1$ to $T$: $T \leftarrow T \cup \{t_1\}$;
Initialize the largest row: $c \leftarrow S_{t_1}$;
Set $S_{t_1} \leftarrow 0$;
**for** $k = 2$ **to** $D$ **do**
 $\quad$ **for** *each token $i$ not in $T$* **do**
 $\quad\quad$ Compute gain: $g_i = \sum_{j=1}^{N} \max(0, S_{i,j} - c_j)$;
 $\quad$ **end**
 $\quad$ Select next token: $t_k \leftarrow \arg\max_{i \notin T} g_i$;
 $\quad$ Add $t_k$ to $T$: $T \leftarrow T \cup \{t_k\}$;
 $\quad$ Update largest row: $c_j \leftarrow \max(c_j, S_{t_k,j})$ for all $j = 1$ to $N$;
 $\quad$ Set $S_{t_k} \leftarrow 0$;
**end**
**return** $T$

---

# B  OVERALL DETAILED ALGORITHM OF TOMA

---

**Algorithm 3:** ToMA with local regions

---

**Input:** Tensor $X \in \mathbb{R}^{B \times N \times d}$ (input tensor), $D$ (number of destinations), $\tau$ (attention
temperature), $F$ (computational layer)

1 $X \leftarrow (X_1, \ldots, X_P)$ ;      /* Reorganize $X$ as local regions */
 where $X_p \in \mathbb{R}^{B \times N_{local} \times d}$ for $p = 1 \ldots P$ and $N_{local} \times P = N$;

2 $D_{local} \leftarrow D/P$; $X \leftarrow X.reshape(B \times P, N_{local}, d)$;

**Step 1: Facility Location**

3 GPU Greedy to get: $(T_1, T_2, \ldots, T_{B \times P}) \leftarrow \text{Greedy}(f_{FL}, D_{local}, X)$;

4 Gather $X_T \leftarrow (X_{1,T_1}, X_{2,T_2}, \ldots, X_{B \times P, T_{B \times P}})$ ;   /* Shape: $B \times P, D_{local}, d$ */

**Step 2: Merge**

5 $A \leftarrow \text{SDPA}(X_T, X, I, \tau)$ ;       /* Shape: $B \times P, D_{local}, N_{local}$ */

6 $\tilde{A} \leftarrow A/A.\text{sum}(-1)$ ;        /* Normalize each row */

7 $X_{merged} \leftarrow \tilde{A}X$ ;     /* Apply Merge, Shape: $B \times P, D_{local}, d$ */

**Computational Layer:**

8 $X' \leftarrow F(X_{\text{merged}}.reshape(B, D, d))$ ;

**Step 3: Unmerge**

9 $X'_{\text{unmerged}} \leftarrow \tilde{A}^{\top} X'$;

10 Group $X'_{\text{unmerged}}$ back to reverse the local region split;

11 **return** $X'_{\text{unmerged}}$

---

## C  MORE ABLATION RESULTS

### C.1  UNMERGE COMPARISON(TRANSPOSE VS PSEUDO-INVERSE)

| Unmerge method | CLIP | DINO | MSE | Time (s) |
|---|---|---|---|---|
| **transpose** | **31.0273** | **0.0569** | 1296.3734 | **4.75** |
| **pseudo-inverse** | 30.9972 | 0.0571 | **1288.2609** | 10.07 |

Table 6: Comparison of unmerge methods (transpose vs. pseudo-inverse) under the condition: 50 recompute steps, ratio=0.5, global attention (globalAttn)

From Tab. 6, we find the difference between transpose and pseudo-inverse method of unmerge shows similar outcome in scores while transpose is significantly faster than pseudo-inverse, we then set transpose as our default setting.

### C.2  SCHEDULE

In this section, we compare the metric of different sharing schedules of destination selections and attention weights computation

| Dst steps | Attn steps | CLIP | DINO | MSE |
|---|---|---|---|---|
| first step | first step | 30.0429 | 0.0773 | 2488.5682 |
| every 10 steps | every 10 steps | 30.8171 | 0.0729 | 1735.4429 |
| every 10 steps | every 5 steps | 30.8652 | 0.0699 | 1632.3441 |
| every 10 steps | every 1 step | **30.9971** | **0.0668** | **1524.6436** |
| every 5 steps | every 5 steps | 30.8923 | 0.0692 | 1608.6099 |
| every 1 step | every 1 step | 30.9196 | 0.0668 | 1551.5814 |

Table 7: Recompute schedule comparison with CLIP, DINO, and MSE metrics

From Tab. 7, we observe that recomputing attention and destination (Dst) steps more frequently generally results in slightly improved performance across the CLIP, DINO, and MSE metrics. Specifically, recomputing attention every step yields the best scores in all metrics, while less frequent recomputation (e.g., every 10 steps) results in slightly worse scores but still competitive results. The difference between recomputation frequencies becomes more noticeable in the MSE metric, where recomputing more frequently leads to lower error values. We select a recomputation schedule of computing attention every 5 steps and destinations every 10 steps because this provides nearly similar performance to the most frequent recomputation (every step) while likely being faster due to reduced computation overhead. This approach strikes a good balance between performance and efficiency.

# D    THEORETICAL COMPLEXITY

We keep necessary constants for the complexity estimate as they are essential factors in practical speedup calculation. Also, we count the total number of multiplications by treating matrix multiplication as multiple dot products, ignoring algorithms with better theoretical complexity. The complexity is $7d^2N + 2dN^2$ for a self-attention block. After using token merge, the complexity is: $(7d^2D + 2dD^2)$ as we reduce the input size from $N$ to $D$. We also define $r := D/N$ as the reduction ratio. Thus, the speedup in terms of the reduction ratio is Speedup $= \frac{7d+2N}{7d\cdot r+2N\cdot r^2}$.

The overhead of submodular optimization is: $N^2d$

The overhead of computing merge attention projection is: $NDd + Nd$

The overhead of merge is: $NDd$

The overhead of unmerge with transpose is: $NDd$

# E    DETAILED RESULTS ON GEMREC & IMAGENET1K

| Method | Ratio | FID | CLIP | DINO | MSE | RTX6000 | V100 | RTX8000 |
|---|---|---|---|---|---|---|---|---|
| baseline_SDXL | 0 | 25.265 | 29.889 | 0.000 | 0.000 | 6.1 | 14.5 | 16.1 |
| ToMe | 0.25 | 25.650 | 29.861 | 0.054 | 1716.131 | 8.7 | 15.0 | 16.9 |
| | 0.5 | 26.726 | 29.712 | 0.071 | 2279.389 | 8.7 | 12.9 | 14.6 |
| | 0.75 | 41.227 | 29.091 | 0.084 | 2344.868 | 8.2 | 11.2 | 12.4 |
| ToMA strip | 0.25 | 25.168 | 29.903 | 0.054 | 1604.185 | 5.6 | 12.6 | 14.5 |
| | 0.5 | 29.110 | 29.524 | 0.074 | 2199.760 | 4.6 | 10.1 | 12.0 |
| | 0.75 | 89.932 | 26.973 | 0.110 | 3185.344 | 4.5 | 8.0 | 9.5 |
| ToMA tile | 0.25 | 25.432 | 29.856 | 0.045 | 1348.644 | 6.2 | 13.6 | 15.7 |
| | 0.5 | 29.192 | 29.629 | 0.063 | 1912.216 | 6.3 | 11.1 | 13.2 |
| | 0.75 | 58.896 | 28.174 | 0.091 | 2802.324 | 6.2 | 9.1 | 10.7 |
| ToMA * | 0.25 | 26.311 | 29.696 | 0.052 | 1866.684 | 5.5 | 12.3 | 13.5 |
| | 0.5 | 38.138 | 29.061 | 0.080 | 3451.150 | 4.9 | 9.7 | 11.5 |
| | 0.75 | 123.366 | 24.963 | 0.106 | 5440.233 | 4.9 | 7.6 | 8.9 |
| ToMA | 0.25 | 25.718 | 29.858 | 0.048 | 1432.562 | 6.0 | 14.3 | 15.9 |
| | 0.5 | 28.875 | 29.640 | 0.068 | 2012.134 | 5.0 | 11.0 | 12.8 |
| | 0.75 | 58.592 | 27.961 | 0.098 | 2785.680 | 4.3 | 8.5 | 9.8 |
| LTB | 0.25 | – | – | – | – | 5.2 | 12.1 | 3.1 |
| | 0.5 | – | – | – | – | 4.0 | 9.9 | 7.8 |
| | 0.75 | – | – | – | – | 3.1 | 8.3 | 6.5 |

Table 8: Comparison of different methods with respect to FID, CLIP, DINO, MSE, and various GPU performance metrics (RTX6000Ada, V100, RTX8000).

# F QUALITATIVE RESULT

## F.1 MORE ON TOMA

Please refer to Fig. 7 for more qualitative result of ToMA.

## F.2 COMPARISON WITH OTHER BASELINE MODELS

Please refer to Fig. 8 for more qualitative result of ToMA and other baseline models.

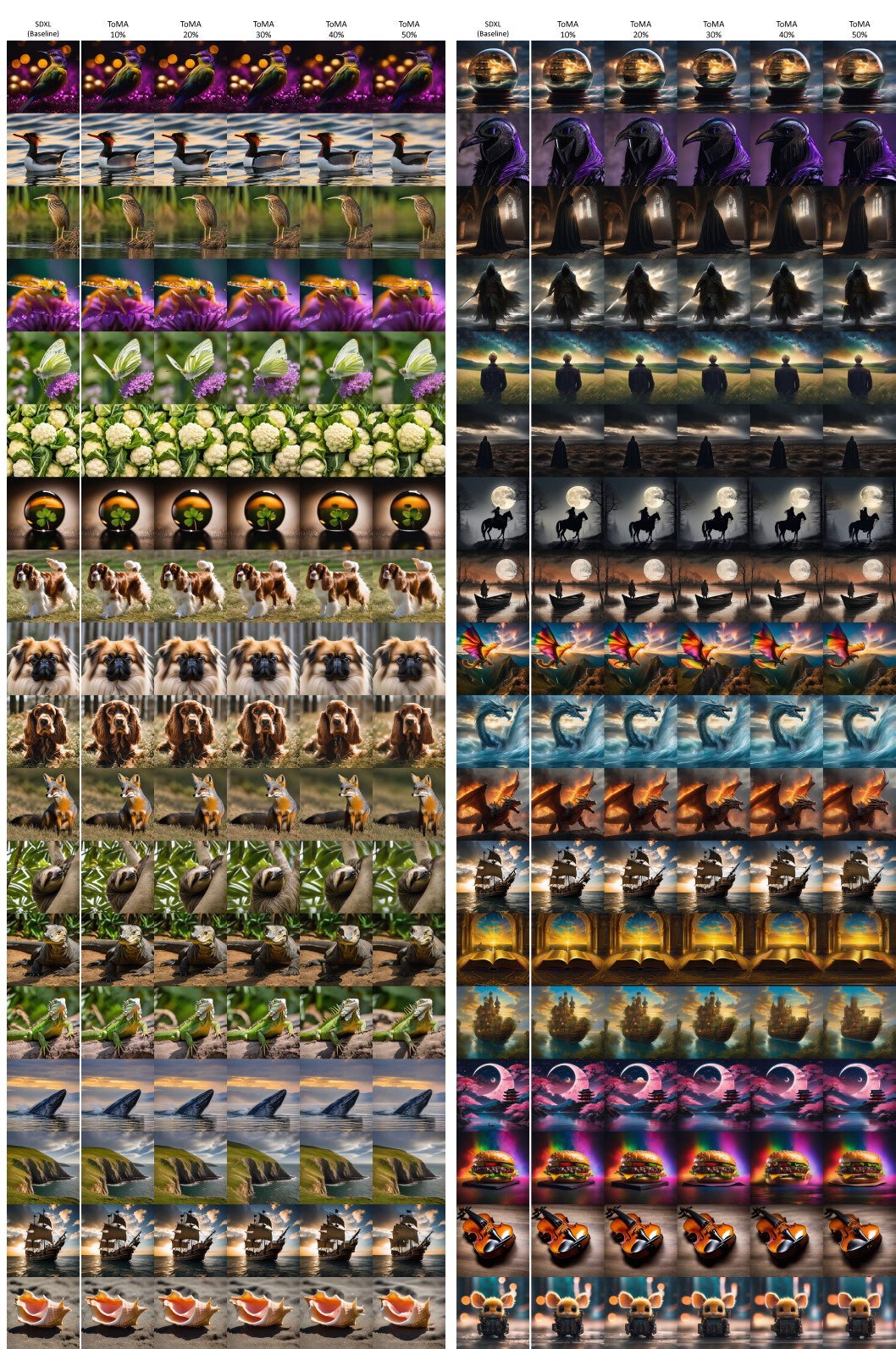

Figure 7: **Visual examples of ToMA**. Even with half of the tokens merged, ToMA consistently preserves image quality and often demonstrates greater robustness compared to other methods (ToDo, ToMeSD).

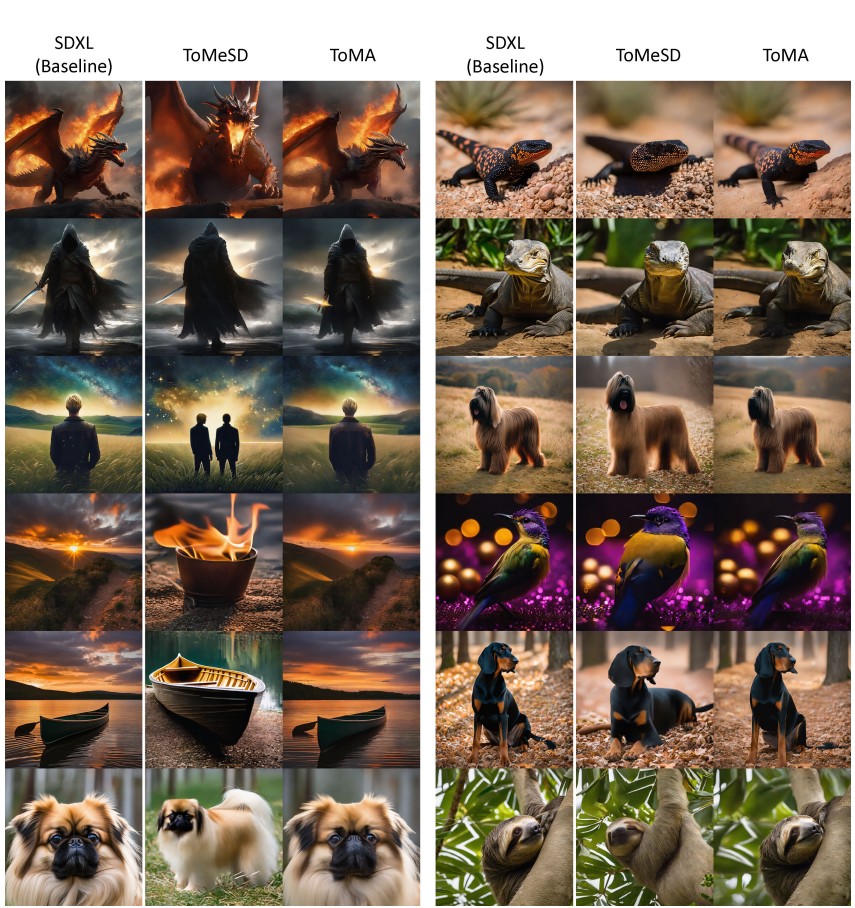

Figure 8: Qualitative comparison between Baseline `SDXL-base`, ToMeSD, and ToMA.

# G   RESULT ON MORE BASELINE MODELS

We have compared ToMA with other baselines(eg. ToMeSD, ToFu, ToDoSmith et al. (2024)) and from the result we find that across all the ratios the ToMA achieve the most speedup and get better image quality compared to ToFu and ToDo.

| Ratio | Method | FID↓ | CLIP↑ | DINO↓ | MSE↓ | Sec/img↓ |
|---|---|---|---|---|---|---|
| **Baseline** | SDXL-base | 25.27 | 29.889 | 0 | 0 | 6.07 |
| 0.25 | ToMA | 25.72 | 29.858 | 0.048 | 1,433 | 6.03 |
|  | ToMeSD | 25.65 | 29.861 | 0.054 | 1,716 | 8.66 |
|  | ToFu | 35.15 | 29.340 | 0.072 | 2,639 | 6.92 |
| 0.50 | ToMA | 28.88 | 29.640 | 0.068 | 2,012 | 5.04 |
|  | ToMeSD | 26.73 | 29.712 | 0.071 | 2,279 | 8.73 |
|  | ToFu | 142.08 | 25.039 | 0.134 | 7,408 | 6.83 |
| 0.75 | ToMA | 58.59 | 27.961 | 0.098 | 2,786 | 4.34 |
|  | ToMeSD | 41.23 | 29.091 | 0.084 | 2,345 | 8.16 |
|  | ToFu | 161.47 | 24.126 | 0.148 | 5,318 | 6.76 |
|  | ToDo | 68.59 | 27.635 | 0.093 | 3,694 | 5.67 |

Table 9: Comparison of SDXL-base and various methods for generating 1024x1024 images for 50 denoising steps. ToDo is given a consistent ratio of 0.75 since it applies a 4x downsample for KV. Metrics are denoted as (↑: higher is better, ↓: lower is better), with the best performance highlighted.

# H  DIFFUSION TRANSFORMERS (DITS)

## H.1  DIT LOCALITY

We examined the hidden states of DiT models, focusing specifically on the `FLUX.1-dev` setting. Using visualization techniques, we analyzed the hidden states at the start of each block and across the denoising timesteps. As shown in Figure 9, the hidden states, despite the lack of convolutional layers, appear to closely represent the true image. Our analysis indicates that this locality is introduced apart from the VAE through the positional embeddings incorporated in DiT models, such as rotary embeddings in Flux and sin/cos embeddings in SD3 and SD3.5. Practically, through our experiments, we applied submodular-based token selection within local regions, which resulted in high-quality images.

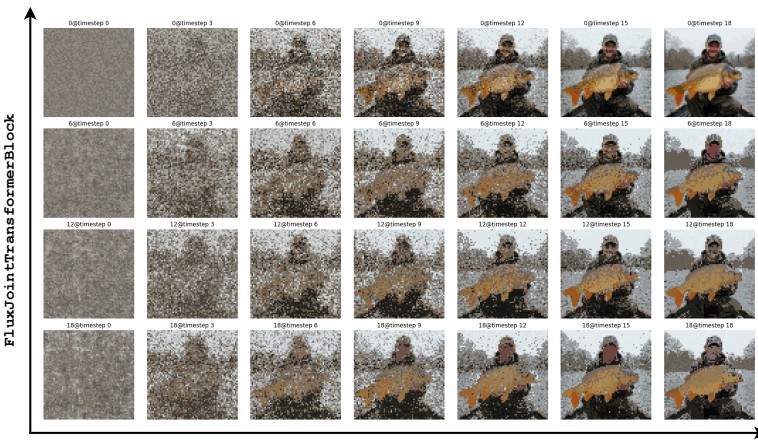

Figure 9: Recolored K-means results on hidden states of Flux.1-dev, across blocks & denoising steps.

## H.2  SPECIAL DESIGN OF TRANSFORMER BLOCKS AND POSITIONAL EMBEDDING

Due to the unique design of the transformer blocks in DiT models, which combine attention blocks and MLPs differently compared to the traditional setup of self-attention, cross-attention, and MLP, existing token merging methods such as ToMeSD, ToFu, and ToDo cannot be directly applied which would lead to all black or pure white noise. Additionally, the influence of positional embeddings further complicates their applicability since the naive application of token merging can lead to the selection of tokens that are not the most similar, which significantly degrades performance.

To address these issues, we implemented specific adaptations to the transformer blocks and positional embeddings, allowing our approach to successfully generate correct images with minimal quality loss which is shown in Tab. FIXME. Our method was selectively skip the first 10 transformer blocks in FLUX.1 to enable better the blend of text and image.

## H.3  RESULTS ON DIT

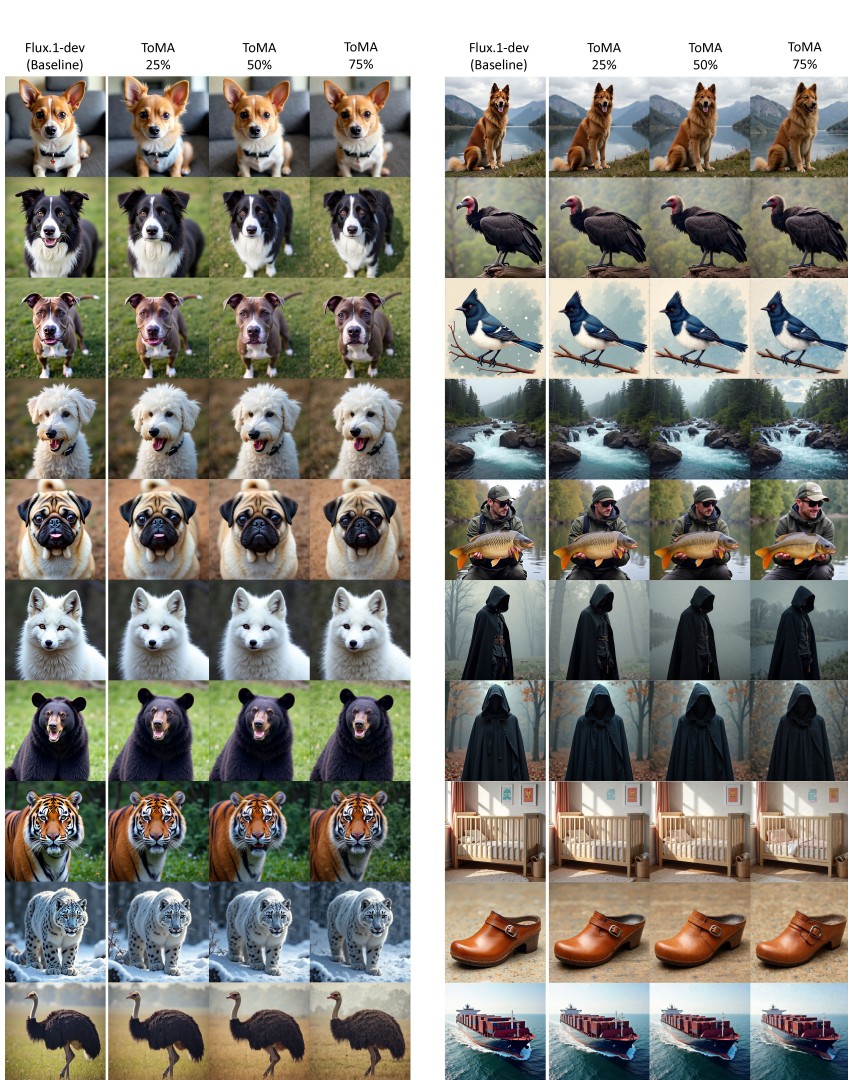

Figure 10: Qualitative comparison between Baseline `Flux1.0-dev` and ToMA.

| Method | Ratio | FID↓ | CLIP↑ | DINO↓ | MSE↓ | Sec/img↓ |
|---|---|---|---|---|---|---|
| **Baseline** | Flux.1-dev | 31.56 | 29.026 | 0 | 0 | 21.03 |
| ToMA | 0.25 | 30.80 | 29.068 | 0.043 | 1,340 | 20.14 |
| | 0.50 | 31.70 | 29.091 | 0.051 | 1,579 | 18.58 |
| | 0.75 | 33.39 | 28.976 | 0.064 | 2,041 | 16.12 |

Table 10: Performance of Flux.1-dev and various methods for generating 1024x1024 images for 35 denoising steps. Metrics are denoted as (↑: higher is better, ↓: lower is better). No other model works on DiT models.

