# OpenReview forum: "ToMA: Token Merging with Attention For Diffusion Models"
_ICLR.cc/2025/Conference — Submitted to ICLR 2025_

### Official Review · Reviewer_c1ky · 2024-10-30

**Soundness:** 4
**Presentation:** 2
**Contribution:** 3
**Rating:** 6
**Confidence:** 2

**Summary:**

Previous token selection methods have overlooked the relationships between tokens and have not utilized the latest attention implementations, limiting actual speedup.
This paper proposes a submodular function-based token selection mechanism and introduces an attention-based approach for merging and unmerging tokens. This design leverages the benefits of modern attention acceleration libraries and is reversible in nature.
As a result, the authors' method achieves an optimal trade-off between performance and efficiency.

**Strengths:**

- The use of attention-based operations for token merging is well-designed and makes sense. Additionally, the authors' choice to make it an invertible function is highly meaningful, and they thoughtfully consider GPU implementations.

- The authors also discuss sharing destination selections across steps, which indeed reduces computational costs and enhances the practicality of the overall approach.

- The experiments are comprehensive and well-executed.

**Weaknesses:**

- The manuscript lacks discussion on the DiT model, focusing only on Stable Diffusion. In the "Local Region" section, it would be beneficial to include insights on how this technology could be adapted for DiT-like models. Without convolution layers, it is unclear if the locality is still evident enough to support the use of this method. Can you provide a brief analysis or discussion on how the locality assumptions might change for DiT models, and whether any modifications to the proposed method would be needed to accommodate those differences.

- Some figures could be improved for better visualization.e.g., it is a little difficult to differentiate different methods in Figure 5.

- The manuscript contains redundant content, specifically in lines L142-L150 and L151-L155, where identical information is repeated.

**Questions:**

My primary concern is the lack of discussion regarding the DiT model. Could the authors provide additional results or discussions specifically related to DiT image generation models, such as PixelArt-Alpha?

---

> ### Author Response · Authors · 2024-11-25
> **Response**
>
> 1. **“*ToMA on DiT*”:**
>
>    Thank you for pointing this out. We’ve added the experimental results for ToMA on FLUX.1 (DiT). Please refer to the general comment for more details. Additionally, we’ve analyzed the locality in the DiT model, and you can find the detailed explanation in Appendix H. We believe the locality of DiT is primarily driven by its positional embedding.
>
> 2. **“*Better Visualization*”:**
>
>    We appreciate your suggestion, and we’ve improved the visualizations as recommended.
>
> 3. **“*Redundant Paragraphs*”:**
>
>    Thank you for the careful review. We’ve revised and removed the redundant paragraphs as suggested.

---

> ### Author Response · Authors · 2024-12-01
> **Response**
>
> Dear Reviewer,
>
> As the end of the discussion period approaches, we are happy to address any additional questions you may have or further clarify your concerns.
>
> Thank you once again for your time and consideration.

---

### Official Review · Reviewer_Py5r · 2024-11-01

**Soundness:** 3
**Presentation:** 2
**Contribution:** 2
**Rating:** 6
**Confidence:** 4

**Summary:**

This paper proposes Token Merge with Attention (ToMA) to tackle the issues of limited image quality due to the loss of important tokens and the inefficiency of attention mechanisms. The authors establish ToMA through three major components: a submodular-based token selection method, an efficient attention implementation, and (un-)merging as (inverse-)linear transformations. Based on this design, the paper significantly reduces the inference time for text-to-image generation model (SDXL).

**Strengths:**

This paper exhibits several strengths:

1.	The motivation and methodology are both reasonable and intuitive.

2.	The generation model (SDXL) is significantly accelerated by merging and unmerging tokens before and after attention, along with additional speed-up settings, all without any loss in quantitative performance indicators.

**Weaknesses:**

This paper exhibits several Weaknesses:

1.	Lack of qualitative comparison with ToMeSD.

2.	The visual effects of ToMA are underrepresented, and quantitative indicators only partially reflect the quality of generation. More samples are needed to substantiate claims about "the best trade-off between image quality and speed."

3.	Current Text-to-Image models (such as Flux and SD3) based on diffusion transformers have achieved new state-of-the-art results. While the paper states that ToMA can be applied to any attention-based T2I model, it is recommended that the authors verify ToMA's performance on the latest T2I models to enhance persuasiveness.

4.	In the bottom of Figure 6, ToMA introduces considerable noise compared to the original result. Does this imply that, despite ToMA showing less performance loss in quantitative evaluations, it incurs greater performance loss in terms of visual perception?

**Questions:**

1.	I noticed that in the smaller steps of Figure 4, the average token intersections across different layers are significantly different. In these steps, could the sharing of both destinations and attention weights between layers lead to a notable loss in performance?

2.	How is the scale of the set of destinations determined? Specifically, how does the size of 𝐷.

3.	The terms "Dino" and "Clip" mentioned in line 475 should be aligned with the entries in Table. 3.

**Details Of Ethics Concerns:**

Nan

---

> ### Author Response · Authors · 2024-11-25
> **Response**
>
> 1. ***“Qualitative Analysis and* Concerns About Noise”:**
>
>     Thanks for the suggestion. We agree that more visuals are necessary to substantiate our contribution. We have added more images and please check out the general comment.
>
> 2. **“*ToMA on DiT*”:**
>
>     We have added our experiment result of ToMA on FLUX.1(DiT). You can check the general comment for details.
>
> 3. ***“Sharing in Small Steps”:***
>
>     Thanks for pointing this out and we have similar observations. However, from a result-wise analysis, which you can refer to in Table. 7 from Appendix C. 2, we find little difference in image quality between our sharing strategy and the non-sharing strategy.
>
> 4. ***“Scale of Destination Set”:***
>
>     In our setting, the size of $D$ is $r\cdot N$ where $r$ is the merge ratio (what percentage of tokens to merge) and $N$ is the original number of tokens. Essentially, the choice of  $D$  depends on the level of speed-up & quality trade-off users desire. For theoretical speed-up in terms of  $D$, please check the appendix D.
>
> 5. ***“Not Aligned Entries”:***
>
>     Thank you for noticing that and we have fixed it.

---

> > ### Comment · Reviewer_Py5r · 2024-11-26
> > **Response to authers**
> >
> > Thanks to the authors for their response.
> >
> > The author's explanations, more visualizations (in SDXL and FLUX) and clearer elaboration of the method (including new method diagrams and pseudo-code) have addressed most of my concerns.
> >
> > Therefore, I upgrade my score.

---

### Official Review · Reviewer_2zks · 2024-11-01

**Soundness:** 2
**Presentation:** 2
**Contribution:** 1
**Rating:** 3
**Confidence:** 5

**Summary:**

This paper introduces Token Merge with Attention (ToMA) to optimize transformer-based diffusion models, addressing inefficiencies in existing token merging methods. By utilizing submodular optimization for token selection, efficient attention mechanisms, and leveraging token locality, ToMA achieves substantial computational speedups with minimal impact on image quality, making it compatible with modern GPU architectures.

**Strengths:**

1. The use of submodular optimization for token selection effectively reduces information loss during merging, improving quality retention compared to previous approaches.

2. The paper's experiments, which utilize metrics such as CLIP, DINO, and FID on high-quality datasets, demonstrate ToMA's balance between efficiency and image quality.

**Weaknesses:**

1. The use of locality to limit the scope of attention for computational efficiency, as implemented in ToMA, is not sufficiently novel. Similar approaches have already been explored in methods such as Sparse Transformer[1], DiffRate[2], ToFu (Token Fusion)[3], making it difficult to assess the unique contribution of ToMA.

2. The experimental comparisons are primarily limited to ToMeSD, without benchmarking against other prevalent methods such as Token Pruning, Flash Attention, DiffRate[2], ToFu[3], and FRDiff[4].

3. The paper is lack of qualitative visual analysis. Without sufficient visual examples, it is challenging to assess ToMA's performance meaningfully, especially in comparison to other acceleration methods.

[1] Child, R., Gray, S., Radford, A., & Sutskever, I. (2019). Generating Long Sequences with Sparse Transformers. arXiv preprint arXiv:1904.10509.

[2] Chen M, Shao W, Xu P, et al. Diffrate: Differentiable compression rate for efficient vision transformers[C]//Proceedings of the IEEE/CVF International Conference on Computer Vision. 2023: 17164-17174.

[3] Kim, M., Gao, S., Hsu, Y.-C., Shen, Y., & Jin, H. (2023). Token Fusion: Bridging the Gap between Token Pruning and Token Merging. arXiv preprint arXiv:2312.01026.

[4] So J, Lee J, Park E. FRDiff: Feature Reuse for Universal Training-free Acceleration of Diffusion Models[J]. arXiv preprint arXiv:2312.03517, 2023.

**Questions:**

Refer to Weaknesses.

---

> ### Author Response · Authors · 2024-11-25
> **Response**
>
> 1. ***“Novelty of Locality”:***
>
>     Thank you for pointing this out. We admit the locality is a common practice in accelerating attention. However, our approach is different from the previous approach. Methods such as Longformer, Sparse Transformers, and Swin Transformer apply locality in attention computation. On the contrary, **ToMA applies locality on the merge operation which keeps the computation attention intact.** Theoretically, we can apply their method on top of our local merge. Our approach is plug-and-play, requiring no retraining, and maintains computational efficiency while still allowing long-range token interactions. Our experiments demonstrate that even marginal elimination of weak attention scores for distant tokens can result in significant image degradation, highlighting the importance of preserving these interactions.
>
> 2. ***“Comparison to other Baselines”:***
>
>    For Flash-attention, there might be a misunderstanding that it is actually orthogonal to our work and we develop ToMA on top of it. It is also a major reason why ToMA performs better compared to ToMeSD or other baselines as their overheads are more notable when utilizing efficient implementation like Flash-attention.
>
>     Thanks for pointing out ToFu, we have added the baseline experiment in Appendix G and please check the general response.
>
>     For Token Pruning, as ToFu is a more advanced version of token pruning, we compare ToMA with ToFu instead.
>
>     For the other two papers, they are not directly related but orthogonal to our work. DiffRate incorporates the compression rate and merges tokens in the vision transformers but in the training stage. FRDiff accelerates diffusion inference by reusing feature maps across time steps but not merging the tokens. But to make the literature review more thorough, we have added them to the related work of the paper.
>
> 3. ***“Qualitative Analysis”:***
>
>     Thanks for pointing out the lack of visual analysis. We have added more images and please check out the general comment.

---

> ### Author Response · Authors · 2024-12-01
> **Response**
>
> Dear Reviewer,
>
> As the end of the discussion period approaches, we are happy to address any additional questions you may have or further clarify your concerns.
>
> Thank you once again for your time and consideration.

---

### Official Review · Reviewer_JdHh · 2024-11-03

**Soundness:** 2
**Presentation:** 3
**Contribution:** 2
**Rating:** 3
**Confidence:** 4

**Summary:**

To address the two main challenges in token merging, this paper introduces the TOMA method. TOMA first uses a submodular-based approach to select diverse tokens for merging. It then leverages efficient attention implementations to minimize merge overhead. By abstracting (un-)merging as (inverse) linear transformations, TOMA enables shared computation across layers and further accelerates processing by operating on tokens within local blocks to exploit image locality.

**Strengths:**

1.	The method uses a submodular function to identify a representative subset of tokens for merging and applies a GPU-efficient vectorized optimization algorithm.
2.	The design of ToMA carefully considers the advantages and limitations of GPU computations.
3.	ToMA achieves 30%-50% speedups without noticeable sacrifice in image quality.

**Weaknesses:**

1. This work seems like an enhanced version of ToMeSD, focusing on updated merge rules and additional locality optimizations, but the contributions may not be substantial enough.
2. Regarding experimental results:
- The paper only tests on the SDXL architecture, limiting generalization claims. As noted in line 372, this method could be extended to SD2 and SD3, so more results on these structures are needed. Actually, using token merging in the DiT structure could theoretically offer greater speedups.
- The results in Table 1 for ToMeSD are strange, as its inference time is longer than the baseline. Were torch and xformer versions verified to match the official implementation during testing? Without a correct ToMeSD implementation, comparisons may lose significance.
- FID scores in Figure 5 exceed 25, unusually high for ImageNet.
- The speedup achieved by ToMA is limited. At a ratio of 0.25, the improvement is just 10%, and while a ratio of 0.75 yields a 20% speedup, it comes with a significant decline in quality metrics.
- The comparison methods are limited; it would be beneficial to include approaches such as “Token Downsampling for Efficient Generation of High-Resolution Images.”
3. Some figures and explanations are unclear, e.g., the X-axis in Figure 5.

**Questions:**

Please refer to the weaknesses part.

---

> ### Author Response · Authors · 2024-11-25
> **Response**
>
> 1. **“*ToMA on DiT*”:**
>
>     Thank you for your suggestion. We provide our result on the FLUX.1(DiT). Please check our general response.
>
> 2. ***“Inconsistent Comparison with ToMeSD”:***
>
>     To our best knowledge, ToMA is the method that has significant speed up with the modern implementation of attention(eg. flash attention/xformers/SDPA).
>
>     The main reason behind the notable inconsistencies is that we utilize a more efficient implementation of attention in the diffusion models of our experiment while the ToMeSD utilizes a much slower version. As a result, ToMeSD’s overhead outweighs its benefits. **This issue was also noted by the ToMeSD author in a related GitHub issue https://github.com/dbolya/tomesd/issues/57.**
>
>     **On the contrary, our method utilize the attention for merging which will continuously benefit from the future more efficient implementation of attention.**
>
>     For more details, we use `AttentionProcessor2_0`, which relies on PyTorch's SDPA kernel with `flash_attention2.0`. This accelerates attention by approximately 10 times compared to the standard PyTorch implementation (as noted by Dao et al., 2023).
>
> 3. ***“FID Score”:***
>
>     In our experiments, the computation of the FID score is implemented using pytorch-fid, with the ImageNet-1k validation statistics provided by heusel2017gans in this h[ttps://github.com/bioinf-jku/TTUR](https://github.com/bioinf-jku/TTUR).
>
>     For the detailed setup, we generated 3,000 images for each configuration using a classifier-free guidance (CFG) scale of 7.5 with class names as prompts corresponding to ImageNet-1k classes. Each prompt was evaluated with three different random seeds. This setup may explain why our FID scores are much higher compared to classifier-guided generation on ImageNet-1k.
>
>     **The scale of our FID scores is also comparable to those reported in the ToMeSD paper**, albeit slightly lower. This difference can be attributed to our adherence to the rule of thumb that the number of images generated for evaluation should exceed the hidden dimension of the last layer of the Inception-v3 model (2048). Specifically, we generated 3,000 images per configuration, whereas ToMeSD only generated 2,000 images per configuration. This ensures a fair comparison and yields more robust results in our case.
>
> 4. ***“Comparison with ToDo”:***
>
>     We have tested “Token Downsampling for Efficient Generation of High-Resolution Images”. In general, ToDo is slower than ToMA and gives lower-quality images. Please check Appendix G for details.
>
> 5. ***“Figure Change”:***
>
>     Thanks for the suggestion, we have changed the X-axis. Please check the paper for a modified version

---

> ### Author Response · Authors · 2024-12-01
> **Response**
>
> Dear Reviewer,
>
> As the end of the discussion period approaches, we are happy to address any additional questions you may have or further clarify your concerns.
>
> Thank you once again for your time and consideration.

---

### Official Review · Reviewer_HfzU · 2024-11-05

**Soundness:** 3
**Presentation:** 3
**Contribution:** 3
**Rating:** 6
**Confidence:** 4

**Summary:**

This paper presents three significant advancements aimed at enhancing the token merging mechanism in diffusion models for generative tasks: identifying representative merge destinations, optimizing the merging and unmerging processes, and reducing computational complexity. Specifically, the proposed method employs a greedy-based algorithm to determine a representative subset that serves as merge destinations. This is followed by an additional cross-attention operation and matrix multiplication to effectively execute the merging process. During the unmerging phase, the authors leverage the inverse (or transpose) matrix from the merging step, thereby improving the overall efficiency of the unmerging procedure. Moreover, the authors introduce strategies to merge only tokens located within the same local region and to share destination and merge matrices across iterations and layers, further mitigating computational costs. When compared to an existing approach (i.e., ToMeSD), the proposed method achieves notable improvements in text-to-image generation tasks across two datasets (GEMRec and ImageNet-1k) evaluated using three metrics (CLIP, DINO, and FID), highlighting its efficacy and substantial contribution to the field.

**Strengths:**

- This paper is well-crafted and effectively articulates both the proposed methodology and the corresponding experimental outcomes.
- The implementation of the approach is methodical and straightforward, which supports practical applicability.
- The comprehensive implementation details, supplemented by the provided code, significantly bolster the reproducibility of the research.

**Weaknesses:**

- The title and scope of the paper may lead to potential misunderstandings. While diffusion models have applications beyond generative tasks, the experiments in this work are solely focused on generation. It would be advisable to revise the title to more accurately reflect the scope of the contributions.
- The experimental evaluation is restricted to text-to-image tasks, which limits the generalizability and perceived practical impact of the proposed approach.
- The discussion and comparative analysis do not sufficiently engage with related work on token merging, such as CrossGET [1] and TRIPS [2], which diminishes the thoroughness of the literature review.
- The comparative evaluation is limited to ToMeSD, and there are notable inconsistencies when compared to the results reported in the original paper.



[1] CrossGET: Cross-Guided Ensemble of Tokens for Accelerating Vision-Language Transformers, ICML 2024

[2] TRIPS: Efficient Vision-and-Language Pre-training with Text-Relevant Image Patch Selection, EMNLP 2022

**Questions:**

- There are numerous existing token merging approaches that extend beyond their application in diffusion models and generative tasks. The proposed method appears to function as a plug-and-play token merging technique. How does it perform when integrated with baseline models and discriminative tasks? Are the improvements consistently observed across these models and tasks?

- Could the authors provide more detailed information on the implementation of the tile-shaped regions?

- The submodular-based destination selection appears analogous to Farthest Point Sampling (FPS). To my understanding, in most 3D applications, the FPS algorithm is implemented with CUDA to achieve acceptable speed. This step seems to contribute significantly to the computational overhead of the proposed method. Could the authors clarify the distinctions between the submodular approach and FPS, particularly in terms of efficiency?

- In the original ToMeSD paper (applied in SD 1.5), the results indicate a reduction in inference time (s/img). However, in Table 1, even at higher compression rates (0.5 and 0.75), this reduction is not evident. Could the authors provide an explanation for this discrepancy?

---

> ### Author Response · Authors · 2024-11-25
> **Response**
>
> 1. ***“Title and Scope Revision”:***
>
>     We have taken your suggestion into account and revised the title to more accurately reflect the focus of our work. Specifically, the revised title highlights the emphasis on generative tasks within diffusion models, ensuring that the scope of our contributions is clear. We sincerely thank you for this valuable suggestion.
>
> 2. ***“Evaluation on Discriminative Tasks”:***
>
>     Our current work primarily aims to enhance the efficiency of token merging specifically within the context of text-to-image (T2I) generation using diffusion models. We acknowledge that token merging is a versatile approach that could be evaluated in a broader set of tasks, including discriminative tasks and integration with baseline models. However, evaluating such applications falls outside the scope of this paper. We consider this as an exciting direction for future work, and we are actively exploring these applications as part of ongoing research.
>
> 3. ***“Related Work on CrossGET and TRIPS”:***
>
>     Thank you for pointing out these two papers for us and we have added these two papers to the related work. CrossGET combines token but on vision-language models with tasks like image captioning and image-text retrieval. TRIPS proposes text-relevant image patch selection but it accelerates the image-language model pertaining. The major difference between these two papers and ToMA is the task. ToMA focuses on image generation of diffusion models and they focus on other image-language tasks like image captioning and pretraining.
>
> 4. **“*Implementation of Tile-Shaped Regions*”:**
>
>     We extract the tile-shaped region like a sliding window without overlapping area. Implementation-wise, it relies on the permute operation or the as_strided operation provided by Pytorch. For more details, please check the code in the supplementary material at the tile_wise_batched_facility function in the facility_location.py.

---

> ### Author Response · Authors · 2024-11-25
> **Response Cond**
>
> 5. **“*Comparison with Farthest Point Sampling (FPS)”:***
>
>     Below, we address the distinctions between our submodular-based destination selection approach and Farthest Point Sampling (FPS) [Eldar et al., 1997] in terms of representativeness, theoretical foundation, and computational efficiency:
>
>     - **Representativeness and Diversity**:
>
>         The submodular-based approach we adopt is fundamentally different from FPS. In FPS, the next point $ v^* $ is selected to maximize the minimum distance from all previously selected points:
>
>         $$
>         v^* = \arg max_{v \in (V \setminus A)} \min_{v' \in A} \text{dist}(v, v'),
>         $$
>
>         where $ \text{dist}(v, v') $ measures the distance (e.g., Euclidean or cosine similarity) between points $ v $ and $ v' $. While FPS selects points to maximize spatial separation, our method ensures both *representativeness* and *diversity* in the selection of merge destinations by optimizing a gain function derived from the facility location problem. Specifically, the greedy algorithm selects the token $ v^* $ from $ V \setminus A $ that maximizes:
>
>         $$
>         v^* = \arg max_{v \in (V \setminus A)} f(v | A),
>         $$
>
>         where $ f(v | A) $ measures the marginal contribution of a new token $ v $ to the representative set $ A $. This ensures that the selected tokens not only capture the most diverse features but also provide a comprehensive representation of the input, outperforming FPS in scenarios with complex feature distributions.
>
>     - **Theoretical Foundations**:
>
>         Unlike FPS, which primarily serves as a heuristic without strong theoretical guarantees, our method is rooted in the theory of submodular optimization, particularly the *facility location problem*. The gain function:
>
>         $$
>         f(v | A) = \sum_{u \in V} \max(0, \text{sim}(u, v) - \max_{v' \in A} \text{sim}(u, v')),
>         $$
>
>         provides a principled framework for selecting tokens.
>
>     - **Efficient GPU Implementation**:
>
>         We recognize that submodular optimization could introduce computational overhead. To address this, we have implemented the gain function computation on GPUs using a highly parallelized design. By vectorizing similarity calculations and marginal gain updates, our method achieves comparable running time to FPS on GPUs while maintaining superior token selection quality. Thus, the computational cost is significantly reduced, making the method practical for large-scale applications.
>
> 6. **“*Discrepancy in Inference Time with ToMeSD*”:**
>
>     To our best knowledge, ToMA is the only method that has significant speed-up with the modern implementation of attention(eg. flash attention/xformers/SDPA).
>
>     The main reason behind the notable inconsistencies is that we utilize a more efficient implementation of attention in the diffusion models of our experiment while the ToMeSD utilizes a much slower version. As a result, ToMeSD’s overhead outweighs its benefits. **This issue was also noted by the ToMeSD author in a related GitHub issue https://github.com/dbolya/tomesd/issues/57.**
>
>     **On the contrary, as ToMA utilizes attention for merging, its speed-up will continue to benefit from any future improvement with the implementation of attention, from which other methods suffer.**
>
>     For more details, we use `AttentionProcessor2_0`, which relies on PyTorch's SDPA kernel with `flash_attention2.0`. This accelerates attention by approximately 10 times compared to the standard PyTorch implementation (as noted by Dao et al., 2023).

---

> > ### Comment · Reviewer_HfzU · 2024-11-27
> >
> > I appreciate the authors' responses, which clarified the unclear implementation details and highlighted the novelty of this work relative to other token merging approaches, addressing most of my concerns. Therefore, I have decided to maintain my score. Regarding the discrepancy in inference time compared to ToMeSD, I understand that the ToMA-only method improves performance by incorporating transformers and flash_attention layers. In this case, it would be helpful to include some theoretical analysis or discussion on the differences between applying the merging process in the xformer architecture versus the original attention architecture. This would strengthen the argument and increase the credibility of the proposed method.

---

### Author Response · Authors · 2024-11-25
**General Response**

*Dear Reviewer*,

We appreciate your insightful comments and the time you've taken to evaluate our work. This is a general response to common questions, and we will reply specifically to each reviewer.

### *More Visualization*

Many reviewers have noted that we do not have enough visualizations to facilitate a qualitative assessment. In response, we have generated 36 groups of additional images at a resolution of 1024x1024, using prompts from the GEMRec and ImageNet 1K datasets. Also, we add more qualitative comparisons between ToMeSD and ToMA. Reviewers can refer to Appendix F for these new visualizations.

### *Additional Baseline Comparison*

We appreciate your feedback regarding the limited set of models we tested. To demonstrate the effectiveness of ToMA, we have expanded our baseline comparisons on SDXL. Specifically, we included two methods—ToFu and ToDo—mentioned by the reviewers. Our results indicate that ToMA is the only method that consistently achieves significant speed improvements while maintaining high image quality. Reviewers can refer to Appendix G for details.

### *ToMA on DiT*

Several reviewers raised concerns about the validity of our approach across different models. To evaluate the generalizability of ToMA, we conducted experiments on FLUX.1-dev.

ToMA on DiT (FLUX.1-dev): We conducted experiments on FLUX.1-dev, which is one of the state-of-the-art DiT models. Due to the unique design of the transformer block with parallel MLP and ROPE, all the existing token merge methods (eg. ToMeSD, ToDo, ToFu, and ToMA) can not be directly applied. We implemented specialized adaptations of ToMA for this model. Our results showed expected acceleration within the transformer block, along with high-quality image generation, even at elevated merge ratios. For the baseline methods, they fail and generate black or white-noised images. For further details, please refer to Appendix H.

---

### Meta-Review · Area_Chair_1R7x · 2024-12-18

**Metareview:**

This work introduced ToMA to address inefficiency issue in existing token merging methods, by utilizing submodular optimization for token selection and leveraging token locality. The effectiveness of approach and the trade-off obtained between efficiency and image quality are appreciated. It received two clear reject and three borderline accept, while being a bit diverged but towards negative. One main concern commonly raised by all reviewers is incompleteness of experiment in many aspects, including limited comparison with prior works, weak baselines, lack of qualitative analysis, etc. Other concerns include similarity with previous works, marginal improvements, etc. The rebuttal, while adding some asked experimental results, didn't convince reviewers to obviously change their opinions. Overall this work, while is of great value, requires more significant work to make it complete to support what it claimed. After considering all the comments and discussions, a decision of reject is made and authors are advised to polish it based on the feedback for the future submission.

**Additional Comments On Reviewer Discussion:**

One main concern commonly raised by all reviewers is incompleteness of experiment in many aspects, including limited comparison with prior works, weak baselines, lack of qualitative analysis, etc. Authors missed some well known prior works mentioned by reviewers. Though authors added several required experimental results, this indicates that this work is far from being complete and ready. One reviewer who gave 6 is with only low confidence score so AC did not put too much weight there. For other four reviewers, the feedback is towards negative and none of them obviously upgrade the score. Therefore AC made the reject decision.

---

### Decision · Program_Chairs · 2025-01-22

Reject